# Neurons for Ejaculation and Factors Affecting Ejaculation

**DOI:** 10.3390/biology11050686

**Published:** 2022-04-29

**Authors:** Kiran Kumar Soni, Han-Seong Jeong, Sujeong Jang

**Affiliations:** Department of Physiology, Chonnam National University Medical School, Hwasun 58128, Korea; jhsjeong@hanmail.net

**Keywords:** ejaculation, spinal ejaculatory generator, lumbar spinothalamic cells, ejaculatory dysfunctions

## Abstract

**Simple Summary:**

Sexual dysfunctions are rarely discussed in our current society. Males experience different sexual dysfunctions, including erectile, infertility, and ejaculatory dysfunctions. In this review only the ejaculatory dysfunction will be discussed. Ejaculation is defined as the ejection of contents collectively from the vas deferens, seminal vesicle, prostate and Cowper’s glands. It is completely controlled by a population of neurons present in the lumbar spinal cord. The presence of lesion in these neurons ceases the ejaculatory behavior in males. This population of neurons was first identified in rats; however, recently it was confirmed that these neurons are present in human males as well. The issues are known as ejaculatory dysfunction. The following are the different types of ejaculatory dysfunctions: early ejaculation, ejaculation into the urinary bladder, late ejaculation and no ejaculation.

**Abstract:**

Ejaculation is a reflex and the last stage of intercourse in male mammals. It consists of two coordinated phases, emission and expulsion. The emission phase consists of secretions from the vas deferens, seminal vesicle, prostate, and Cowper’s gland. Once these contents reach the posterior urethra, movement of the contents becomes inevitable, followed by the expulsion phase. The urogenital organs are synchronized during this complete event. The L3–L4 (lumbar) segment, the spinal cord region responsible for ejaculation, nerve cell bodies, also called lumbar spinothalamic (LSt) cells, which are denoted as spinal ejaculation generators or lumbar spinothalamic cells [Lst]. Lst cells activation causes ejaculation. These Lst cells coordinate with [autonomic] parasympathetic and sympathetic assistance in ejaculation. The presence of a spinal ejaculatory generator has recently been confirmed in humans. Different types of ejaculatory dysfunction in humans include premature ejaculation (PE), retrograde ejaculation (RE), delayed ejaculation (DE), and anejaculation (AE). The most common form of ejaculatory dysfunction studied is premature ejaculation. The least common forms of ejaculation studied are delayed ejaculation and anejaculation. Despite the confirmation of Lst in humans, there is insufficient research on animals mimicking human ejaculatory dysfunction.

## 1. Introduction

Ejaculation is a spinal reflex. It is the forceful ejection of seminal fluid by the males at the end of coitus [1,2]. During ejaculation, the prostatic urethra has pressure of more than 5 m water pressure [3,4]. Specifically, the parasympathetic nervous system maintains the heart rate at relaxed levels; however, the sympathetic nervous system increases the rate of heart rate through the secretion of epinephrine. At the time of ejaculation, the heart rate may increase by 100% in males owing to sympathetic effects [5,6,7]. In rats, ejaculation is associated with reward [8]. It should not be confused with climax or orgasm, which provides pleasurable feelings. Orgasm and ejaculation can be considered a single incident, however, both are different biological processes [9]. Orgasm can be absent in some men with ejection of seminal contents during ejaculation [10], whereas in some men, ejaculation is absent, and they experience intense orgasm [11,12]. 

Ejaculation delivers sperm to the female genital tract for offspring generation. The distal epididymis, vas deferens, seminal vesicle, prostate, prostatic urethra, and bladder neck are the male reproductive organs involved in ejaculation [2,13]. The duration and degree of ejaculation vary remarkably among men and individuals in different situations [14]. In rats, ejaculation is distinguished by a prolonged, intense force (750–2000 ms), and much slower dismount [15]. This review aims to explain the neural regulation of ejaculation, its abnormalities, and its etiology. Moreover, this review discusses ejaculation physiology and abnormalities in animals and their correlation with humans, as well as ways to improve studies that allow mimicking this pathology in humans. 

## 2. Neurophysiology of Ejaculation

Ejaculation remains in the control of the parasympathetic (sacral) and sympathetic (thoracic) autonomic nervous systems and spinal centers [16]. It remains intact in animals with complete spinal cord transection, which provides evidence that ejaculation is controlled at the spinal level, despite the loss in connection from supraspinal regions [17]. The integration between spinal centers and the autonomic nervous system is organized by interneurons that shape the spinal ejaculation generator [SEG] [18]. Based on a detailed study of the spinal gray matter and its cellular composition under a microscope, the gray matter of the spinal cord was found to be split into 10 laminae [I–X] [19]. The parasympathetic nucleus (sacral) innervates the prostate and seminal vesicles located in the S2–S4 segments of lamina VII [16]. The sympathetic nucleus (thoracic) innervates the smooth involuntary muscles of the seminal tract and bladder neck located in the T12-L2 segments of lamina VII [20,21]. Motor neurons that govern the pelvic-perineal striated muscles remain in the Onuf’s nucleus of the ventral horn of the segments [22]. The spinal ejaculation generator or lumbar spinothalamic cells are located in lamina X and the medial part of lamina VII of the gray matter in the lumbar L3–L4 spinal cord of rats [23,24]. They are called lumbar spinothalamic cells as they have connections in the lumbar spinal cord and thalamus [22]. Recently, it was confirmed that the spinal ejaculation generator in male humans is in the L3–L5 segment [16]. 

The marker for neural activation is an increased level of Fos. The activation of these lumbar spinothalamic cells is triggered by stimuli associated with ejaculation; however, mounts or intromissions do not trigger Fos expression in lumbar spinothalamic cells (Figure 1). Activation of lumbar spinothalamic cells or spinal ejaculation generators causes ejaculation [25]. Injury to these neurons acutely compromises ejaculation; hence, these neurons, besides carrying ejaculation-specific sensory information to the brain, also trigger ejaculation [25]. These lumbar spinothalamic cells transport sexual information that is sensory cues prior ejaculation to the thalamus [26] and consolidate the details of the neural interconnection between the somatic/autonomic centers in lumbar spinothalamic cells and the spinal cord [27,28,29]. Interneurons of the spinal ejaculatory generator contain galanin [30], cholecystokinin [31,32], encephalin [33], and gastrin-releasing peptide [34,35,36]; they also co-express androgen receptors [37] and the substance P receptor (neurokinin-1 receptor) [23]. 

### 2.1. Neurons for Ejaculation in Non-Mammalians: Corazonin

In non-mammalians, mostly arthropods, there is an 11 amino acid neuropeptides known as Corazonin neurons [38]. Corazonin neurons are specifically located in abdominal ganglion [39]. It is equivalent to the Gonadotrophin Releasing Hormone that is essential for regulation of growth, ethanol-related behavioral states, stress responses and ejaculation. Activation of corazonin neurons induce ejaculation by interacting with Neuropeptide F (*npf)* that is rewarding to *Drosophila* males. However, stimulation of corazonin neurons was enough for a rise in *npf* level, which is similar to a post-copulation state where copulation is without an actual female [40,41]. Stimulation of corazonin neurons to induce ejaculation is possible by “fictive mating”, that is by opto- and thermogenetic [41,42].

Ejaculation consists of two phases: **emission** and **expulsion** [13,16,43,44].

### 2.2. Emission

The beginning of this phase involves the shutdown of the bladder neck to check for retrograde ejaculation [45]. It is followed by a mixture of seminal vesicle, prostatic, vas deferens, and Cowper’s gland secretions into the prostatic urethra [46]. The initiation of the emission phase of ejaculation is not under individual will or controlled cerebrally and can be evoked through visual erotic stimulation or physical stimulation [47,48]. The organs involved in this phase obtain deep autonomic innervations formed by the sympathetic and parasympathetic nerves from the pelvic plexus [48]. The basic neurotransmitter required in the stimulation of the sympathetic nervous system is norepinephrine, and this is balanced by acetylcholine, the parasympathetic neurotransmitter. Sympathetic nerves at levels T10–L2 that leave the spinal cord commence peristaltic contraction of the prostate smooth muscle, seminal vesicles, vas deferens, and epididymis [49]. Once the semen reaches the posterior urethra, ejection of the semen and its content becomes unavoidable [48].

### 2.3. Expulsion [Anterograde Ejaculation]

Expulsion follows emission, where semen is pushed out as the consequence of the rhythmic contractions of the striated muscles of the pelvis and the ischiocavernosus, bulbospongiosus, and perineal muscles [43]. Electromyographic [EMG] studies of the bulbocavernosus or bulbospongiosus have reported evidence of ejaculation or expulsion in animals following electrical or mechanical stimulation of genital structures or of the dorsal nerve of the penis [32,50], which is controlled by lumbar spinothalamic cells. This induces ejaculation by consolidating the sensory information conveyed by the dorsal nerve of the penis, which is the sensory branch of the pudendal nerve [34]. The pudendal nerve, which begins at the S2–S4 level of the sacral spinal cord, causes rhythmic involuntary contractions (Figure 2) [49]. The ejaculation response in men is generally approximately 10–15 contractions [51]. The intra-seminal vesicle pressure during expulsion in rats is approximately 61.4 mmHg [52]. Men cannot undergo a series of ejaculations rapidly; men enter a refractory period immediately after ejaculation; it is a regaining time in which no ejaculation is possible. The refractory period time varies among individuals, ranging from a few minutes to hours [53].

## 3. Ejaculatory Dysfunction

Ejaculatory dysfunction can be classified into four types: **premature ejaculation**, **retrograde ejaculation**, **delayed ejaculation**, and **anejaculation** [54]. Here, delayed ejaculation and anejaculation have been discussed together because the most severe form of delayed ejaculation is anejaculation. Almost all factors for anejaculation are extreme conditions of delayed ejaculation (Figure 3) [20,55,56]. Premature ejaculation and retrograde ejaculation have been widely studied [57]. Unfortunately, delayed ejaculation and anejaculation are the minimally studied ejaculatory dysfunctions [58,59]. 

## 4. Premature Ejaculation

Premature ejaculation [PE] is the most common type of ejaculation dysfunction in men, with a prevalence of up to 75% [48,60]. Premature ejaculation [PE] is a male sexual dysfunction identified by ejaculation that always or nearly always occurs prior to or within 1 min of vaginal penetration [61]. Female partners of men with premature ejaculation [PE] were reported to have worse relationships compared to men without premature ejaculation [PE] [62]. 

Some attempts have been made in rats to create a premature ejaculation model for detailed study. Based on the number of ejaculations in a 30 min period, a premature and delayed ejaculatory animal model has been proposed [63]. A pharmacological model for premature ejaculation has been developed by administering selective 5-HT1A receptor agonists, including flesinoxan, FG-5893, and 8- OH-DPAT [63,64,65]. These drugs reduce ejaculation time, intromission, and mount frequency; however, the actual mechanism of action of these drugs is not clear.

Waldinger suggested four subtypes of PE: [1] lifelong, [2] acquired, [3] natural variable, and [4] premature-like ejaculatory dysfunction [66]. Lifelong PE, which affects men who have never attained ejaculatory control, but have no erectile or desire difficulties; acquired PE, which affects elderly men and is associated with erectile difficulties; natural variable PE is a normal variation in sexual functioning in which a man will have normal and PE at different periods of time; premature-like ejaculatory dysfunction describes men who present with ejaculatory functioning within the normal range, but their PE occurs due to misunderstandings or partner factors [67]. 

**ETIOPATHOGENESIS OF PREMATURE EJACULATION: *Psychological*** and ***biological*** factors have conventionally been responsible for this [68]. 

### 4.1. Psychological 

Factors include anxiety, depression, guilt, stress, history of sexual suppression, lack of poor body image, sexual abuse, problems in understanding among partners, and early sexual experience [43,48,69,70,71,72]. Anxiety is considered the primary cause of rapid ejaculation. The sympathetic nervous system increases anxiety levels and is responsible for rapid ejaculation, whereas low anxiety delays ejaculation [48]. Depression is significantly correlated with decreased orgasm in patients suffering from myasthenia gravis [73]. Cortisol is a hormone associated with mental stress related to high alertness in stressful situations. A recent study showed a relationship between cortisol, stress, and premature ejaculation [74]. 

### 4.2. Biological

Factors include endocrine, genetic, urological, and others [48,69].

**The thyroid** gland is also an endocrine factor considered in sexual disorders. It has been reported that most animals [75] and patients with thyroid hormone disorders experience sexual dysfunction, such as PE, which can be changed by normalizing thyroid hormone levels [76]. Premature ejaculation is observed in patients with thyroid disorders. However, it decreased from 50% to 15% after 2–4 months of treatment [76]. The hyperthyroid rat ejaculation time was shorter than that of the control and returned to normal after treatment [75]. Excessive thyroid hormone levels cause premature ejaculation, as these are clinically related. A high level of thyroid hormones should be regarded as a reversible and novel causative risk factor for premature ejaculation [77,78]. 

Thyroid hormone levels and ejaculatory time duration have shown an inverse relationship in different reports. Hyperthyroidism is associated with shorter ejaculatory time or PE, whereas hypothyroidism is strongly accompanied by longer ejaculatory time or delayed ejaculation [76,79]. Previous patient data showed the widespread presence of PE in patients with hyperthyroidism, which was approximately 42.4%. In the aforementioned study, it was observed that thyroid-stimulating hormone [TSH] levels altered ejaculation latency, independent of age and testosterone level. Furthermore, the findings showed that hikes in TSH levels also caused increases in intravaginal ejaculatory delay time [IELT] levels [79]. A study involving 94 healthy men and 107 men with PE concluded that free T4 levels were notably higher in the PE group than in healthy controls [80]. In another case–control study involving 39 control men and 63 men with PE, TSH levels were notably lower in men with PE, however, no significant changes were found in their free T3 or free T4 levels [81]. Based on this evidence, it has been suggested that the connection between hyperthyroidism and PE may be secondary to higher sympathetic activity [76].

**Testosterone** is considered the principal hormone involved in male gonad formation and ejaculation control. However, there are some contradictory data regarding the correlation between testosterone levels and PE [69]. In a study between men with and without PE, there were no notable differences in the levels of gonadal hormones (luteinizing hormone and free and total testosterone) [82]. In contrast, young patients with PE have been reported to have higher total and free serum testosterone levels. It has been proposed that testosterone plays an excitatory role in ejaculatory control [83]. Another study showed that follicle-stimulating hormones and free testosterone in the serum were elevated in patients who had earlier PE compared with control men [84]. Based on a comprehensive review, there are contradictory results regarding the correlation between PE and testosterone. Therefore, more and larger studies are needed to better understand the relationship between PE and testosterone.

**Diabetes mellitus** (DM) is another common disease. Diabetes mellitus [DM] is a disorder in which there is a markedly higher level of blood glucose due to insufficient insulin, β-cell dysfunction, insulin resistance, or both [85]. Many animal experiments and human records have shown ejaculatory disorders in diabetes subjects. Due to the differences between normal ejaculation, PE, and anejaculation, it is difficult to explain ejaculatory dysfunction in animals. Many factors affect ejaculation in animals, such as animal type, models formed by different techniques, lab experimental plans, and the number of animals used. Some results show variations in the results of DM model animal experiments. Some experiments showed that there was no change in sexual performance in the control and DM animal models [86,87]. A long ejaculation time or delayed ejaculation has been shown in some animal models [88,89,90]. No ejaculation or anejaculation has been observed in some animal models [91,92]. Some animal models have shown reduced PE ejaculation times [93,94]. These findings suggest that there are various types of ejaculatory dysfunction in animal models. Early insulin replacement has been shown to control seminal emission, suggesting that insulin can play a role in preventing ejaculatory dysfunction. Long-term exposure to glucose may cause permanent ejaculatory dysfunction, which shows that once the dysfunction or the issue begins, late insulin therapy cannot recover normal ejaculation function [95]. Animal studies have shown that hyperglycemia is effective in amending the contractility of the epididymis, vas deferens, seminal vesicles, prostate, bladder neck, and urethra by modulating neurotransmitter release [96,97]. The pathology of ejaculatory dysfunction caused by diabetes can be understood by some experiments that show the effects of experimental diabetes on the emission of semen. Chronically, streptozotocin-diabetic animals showed decreased reaction to stimulation of the sympathetic supply of the vas deferens, which may be due to degenerative changes in the autonomic nervous system [96,98,99]. Moreover, reactive oxygen species (ROS) may be the reason for the reduced sympathetic neurotransmission and unusual function of diabetic vas deferens in streptozotocin-induced diabetic animals [99,100]. Changes in serotonin receptors 5-HT also impair serotonergic transmission to the rat brain in animals with long-term hyperglycemia [101,102]. PE is associated with reduced serotonin neurotransmission [103]. However, the mechanisms by which diabetes causes ejaculatory dysfunction remain unclear. Ejaculatory dysfunction occurs in 40% of men with diabetes [104]. PE was higher in men with diabetes than in those without [78.8% vs. 47.5%, *p* = 0.001], which indicates the prevalence of PE in individuals with diabetes [105]. ED [erectile dysfunction] is the primary cause of PE in type 2 DM. With this in mind, one study showed that ED was reported in 95% of patients with type 2 diabetes. Furthermore the study also reported that males with a diabetes history of ≥10 years have a 2.7 fold greater likelihood of PE than those with a history of <5 years [106].

**Obesity and metabolic syndrome** are characterized by a lack of outdoor activities and aging [107,108]. Patients with metabolic syndrome showed a higher prevalence of PE and higher waist circumference, (35.2%) and (51%), respectively, than the control groups (7.6%) and (24%) [109,110]. The mechanism underlying the relationship between metabolic syndrome and PE is not yet fully understood. Some reports have shown that the cause may be depression, since it is known that depression can cause PE and metabolic syndrome [111,112].

**Vitamin D** is a steroid hormone produced in the skin. It is produced by exposure to sunlight [113]. Administration of 2.5 mg of vitamin D3 completely prevented male rat ejaculation [114]. Vitamin D causes anxiety, which may be a probable cause of PE in vitamin D deficiency patients [115,116]. Vitamin D supplementation after 6 months showed improvement in anxiety symptoms [117].

**Genetic factor** studies indicate that 5-HT1a receptor gene polymorphisms, the 5-HT transporter gene-linked polymorphic region (5-HTTLPR), and 5-HT2c receptor gene polymorphisms may be involved in the progression of PE [118,119]. Lifelong PE has been genetically determined in some men [120].

**Urological factors** are part of the urological organs involved in ejaculation (Figure 2). It can be understood that inflammation or disease in these organs may be a cause of ejaculatory dysfunction. Patients with chronic prostatitis may experience PE [121]. The exact pathology connecting prostatitis and PE has not yet been elucidated. However, some researchers have proposed that inflammation of the prostate may lead to changes in the regulation and sensation of the ejaculation reflex via a neurophysiologic pathway [122]. 

**Other** factors include **low seminal plasma magnesium levels** and significantly decreased levels of magnesium in the seminal plasma in PE patients [123]. Lower seminal plasma magnesium levels can cause an increase in thromboxane A2 levels, which in turn causes an increase in endothelial intracellular calcium and a decrease in nitric oxide levels. A decline in nitric oxide levels can cause contraction of the penile muscle, leading to PE [124,125].

**Varicocele** engorges the testicular veins in men. It is also considered one of the causes of male infertility [126]. Additionally, it is associated with PE in patients [127,128]. The exact mechanism of varicocele and PE is difficult to explain; however, it is speculated that varicocele causes intrapelvic congestion, which causes prostatitis or prostate inflammation. Further protection may lead to changes in the regulation and sensation of the reflex of ejaculation via a neurophysiological pathway [122,129].

## 5. Retrograde Ejaculation

We have already previously described that ejaculation is a reflex. Retrograde ejaculation causes ejaculatory dysfunction, wherein partial or complete passage of semen enters the urinary bladder rather than passing out through the urethra, which occurs due to complete contraction inhibition of the bladder neck [2]. This effect is due to nerve pathology in the sympathetic nervous system or due to biological injury [130,131]. Approximately 0.3% to 2% of infertility in men is due to RE [132]. It occurs when little or no semen is ejaculated, however, the sensation of orgasm and contraction can be felt at the base of the penis in the ischeo- and bulbo-cavernosus muscles. Patients complain of dry ejaculation if no semen is ejaculated, followed by ‘‘white urine’’ when emptying the urinary bladder. Detection of sperm in the urine is a confirmed diagnostic test for RE [130]. 

**ETIOPATHOGENESIS OF RETROGRADE EJACULATION:** These factors are classified as ***pharmacological***,***neurogenic*** and ***other*** causes of retrograde ejaculation [133].

### 5.1. Pharmacologic Factors

RE can be induced in one individual undergoing treatment for lower urinary tract symptoms using an alpha-receptor antagonist. Other drugs used for the treatment of hypertension, antidepressants, and antipsychotics have similar effects. Since it is understood that the sympathetic nerves control the contraction of the bladder neck, any drug that obstructs the shutdown of the bladder neck can lead to partial contraction of the bladder neck and cause RE [14,133,134].

### 5.2. Neurogenic Factors

Include nerve injury or damage. Spinal cord injury is considered a major neurological factor for RE [133]. In patients with spinal cord injury [SCI], sexual function is one of the top priorities for a good quality of life [135,136]. Other neurogenic factors include multiple sclerosis, myelodysplasia, and the most common type of uncontrolled diabetes that causes neuropathy [137]. Diabetic RE patients have lower intraureteral pressure than healthy individuals with diabetes who have somatic innervation of the outer ureteral sphincter [138]. RE is more regularly linked to the composition of the internal urethral sphincter at the time of prostatectomy or after retroperitoneal lymph node dissection [RPLND] [133]. 

### 5.3. Other Factors

BPH (benign prostatic hyperplasia) is the enlargement of prostate gland in elderly male population causing lower urinary tract symptoms (LUTS). There are many evidences showing BPH surgery, one of the causative factor for retrograde ejaculation [139].

## 6. Delayed Ejaculation and Anejaculation

These are unusual forms of male ejaculatory dysfunction, indicated by a noticeable delay in ejaculation or an inability to achieve ejaculation [55,140]. These are the least commonly studied forms of male ejaculatory dysfunction, with an approximate prevalence of 1–4% in males [141]. Difficulties in defining DE are associated with the fact that orgasm and ejaculations generally occur simultaneously, despite being two different phenomena [140]. An animal model for DE/AE has been developed, as previously described in the PE section [63]. An important clinical feature seen in DE is that men are incapable of ejecting semen during intercourse with a partner, despite reaching orgasm and ejaculating during solo masturbation [141].


**ETIOPATHOGENESIS OF DELAYED EJACULATION AND ANEJACULATION:**


These factors are ***psychological*** and ***biological***.

### 6.1. Psychological Factors Include Religious Factors, Insufficient Arousal, Masturbation, and Homosexuality

**Insufficient arousal:** Actual subjective arousal is absent in males, whereas sufficient erection is present; however, orgasm is absent. This type of insufficient arousal causes DE/AE [142]. Drugs that treat erectile dysfunction also cause insufficient arousal and absence of orgasm [140]. 

**Masturbation:** This is also a cause of DE/AE. Men who masturbate to achieve sexual satisfaction have individual durations, pressures, speeds, and strengths required to produce an orgasm, and these can differ from those required with a partner. As a result, they consider it impossible or difficult to attain orgasm with their partners [143].

**Homosexuality:** Men in homosexual situations are said to have a highly elevated prevalence of DE compared to heterosexual men, however this finding is not so precise due to very little work done on it [144,145].

### 6.2. Biological Factors Include Age, Race, Genetic, Congenital, Endocrine, Neurogenic, Infection/Inflammation, and Pharmacological Factors

**Age**: This may be linked to a decrease in the sensitivity of the penis, which is due to loss of penile receptors and sensory axons [146,147].

**Congenital**: The female reproductive organs, such as the oviduct and uterus, are formed by the Müllerian duct during the gestational period. The Wolffian duct gives rise to the kidneys and male reproductive organs [148]. Any inborn deformity of the Wolffian duct or partial remnant of the Müllerian ducts may cause DE/AE [149].

**Genetic**: Patients taking serotonergic antidepressants have been reported to have sexual dysfunction. There is higher risk of DE/AE in patients who take selective serotonin reuptake inhibitors (SSRI) [150]. Genomic methods can be used to identify genes and possible genetic risk factors for SSRI-induced DE [151]. 

**Neurogenic**: Men with multiple sclerosis report DE/AE [152]. In addition, the potential to ejaculate is compromised by SCI. The ejaculation rate was higher in patients with lower motor neuron lesions (15%) than in those with upper motor neuron lesions (5%). The feeling of orgasm or pleasure may be absent or completely lost in patients who are able to successfully ejaculate, which can also cause DE [47]. Retroperitoneal lymph node dissection (RPLND) and bladder neck surgery are also causes of DE [153].

Urinary tract infection, pelvic inflammation, and chronic prostatitis are also causes of DE/AE [154].

**Endocrine**: A reverse connection has been revealed between thyroid hormone levels and ejaculatory duration. Hypothyroidism is robustly linked to a longer ejaculatory duration in DE/AE [79]. Older men with reduced total testosterone and free testosterone levels showed a higher incidence of DE [83].

**Pharmacological**: Antidepressant drugs composed of serotonin also cause DE/AE. A seven-fold risk of DE/AE can be observed in selective serotonin reuptake inhibitor (SSRI) users [150]. 

**Prostate cancer:** It is the third most common type of cancer diagnosed [155]. Radiation therapy and prostatectomy is the method of treatment for prostate cancer. More than 89% patients reported lack of ejaculation after radiation therapy [156]. Anejaculation is anticipated following prostatectomy because prostate and seminal vesicles are removed [157]. 

## 7. Future Directions and Limitations

Male ejaculation and ejaculatory dysfunction remains mostly unexplored topic in the field of scientific study. Spinal ejaculatory generator (SEG) or lumbar spinothalamic neurons offer a highly compliant tool for studying different types of ejaculatory dysfunction. Ejaculation in lab animals can be studied in three different ways. 

First is manually allowing the opposite gender animals to mate. Ejaculation in males shows activation of Lst neurons, which can be traced by increased Fos level immediately after ejaculation [25]. Fos level can be examined by immunohistochemistry. Second is mechanical or electrical stimulation of genital structures or dorsal penile nerve, respectively. Stimulation causes contraction of bulbocavernosus or bulbospongiosus muscles. This can be traced by electromyographic study [32,50]. Third is optogenetics, where light is used to stimulate the nerve cells [158]. Optogenetics stimulating ejaculation neurons are also suggested in non-mammalians species [41,42]. 

Previous attempts explained the use of male rats to create premature and delayed ejaculation models by number of ejaculations in a 30 min period, usage of drugs in pharmacological models of premature and delayed ejaculation [63,64,65]. There is a lack of sufficient findings with these animal models in relation to spinal ejaculatory generator. Pharmacological and biological factors affecting retrograde ejaculation are summarized in this review. Spinal cord injury can be considered as one of the ideal models in animal research because spinal cord injury (SCI) is a devastating neurological disorder, which affects around 250,000 to 500,000 individuals each year [159]. Only 9% of men with SCI can ejaculate by masturbation [160], whereas penile vibratory stimulation or electroejaculation is required by a large population of men with SCI [161,162]. Spinal cord injury model in rats have been used to collect partial ejaculation recovery after infusion of dopamine agonist [163]. However, it is not clear whether it is anterograde ejaculation or retrograde ejaculation. 

Limitations of this studies are Fos tracing in spinal cord is a terminal experiment, so we will not be able to get further findings for a second attempt. Frequent stimulation of dorsal penile nerve may not respond to stimulus because after ejaculation, the animal has refractory period during which the response of stimulus cannot be seen [25]. Another limitation is within the animal model, diabetes causes premature ejaculation and also retrograde ejaculation in humans, therefore it may be difficult in finding among animals if they show premature or retrograde ejaculation. Similar is the case with SCI model, where animals can show retrograde or anejaculation in laboratory findings.

## 8. Conclusions

The identification of a potential spinal ejaculation generator is an important breakthrough in the field of sexual function. Several studies have been conducted to understand the regulation of ejaculation after the identification of a spinal ejaculation generator in rats [23]. Furthermore, recent findings have confirmed the presence of a spinal ejaculation generator in humans [16]. Ejaculation is a complicated process that includes different anatomical and neural compositions as well as large-scale neurochemical and hormonal regulation. As evidenced in this review, sexual problems are one of the major issues in relationships between men and women in our society. It has not been addressed properly, which may be due to the lack of interconnection between human ejaculatory dysfunction and the spinal ejaculatory generator. A detailed study on the different ejaculatory dysfunction types should be conducted in animal models, with the spinal ejaculation generator or lumbar spinothalamic cells at the center of the experiment. The spinal ejaculation generator or lumbar spinothalamic cells present in the L3–L4 region and their activation and inhibition at different time intervals may be the best mechanisms to study in animal models. Ejaculation in animal models can also be predicted by contraction of the bulbocavernosus muscle. Thus, knowledge about spinal ejaculation generators might be useful in the production of different animal models for the different types of ejaculatory dysfunctions discussed in this review.

## Figures and Tables

**Figure 1 biology-11-00686-f001:**
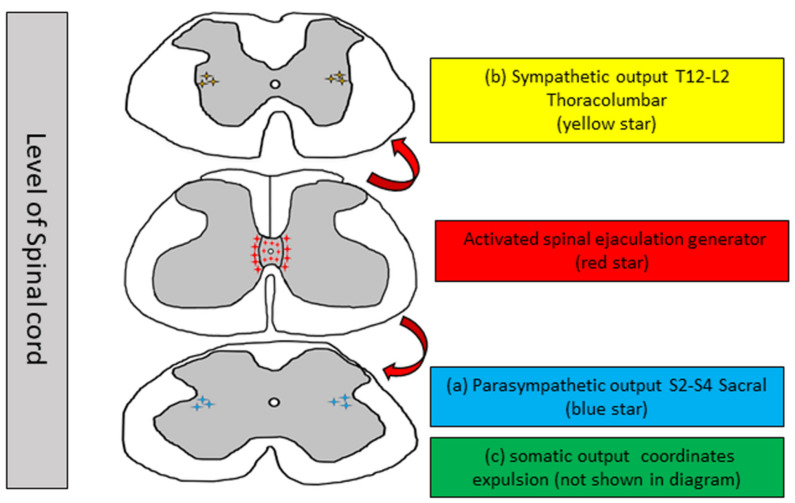
Schematic diagram of level of spinal cord and neurons regulating ejaculation. Activated spinal ejaculation generator directs and synchronizes the activity of: (**a**) parasympathetic output that innervate prostate and seminal vesicles secreting seminal fluid. (**b**) Sympathetic output that innervate smooth muscle cells of the seminal tract and the bladder neck. Contraction of the seminal tract accumulate spermatozoa that is mixed with the seminal fluid to the prostatic urethra. The neck of bladder remains closed to prevent retrograde ejaculation. (**c**) Somatic output that innervate the pelvic striated muscles (not shown in this figure). The external urethral sphincter relaxes and rhythmic contractions of the bulbospongiosus and ischiocavernosus muscles are responsible for rhythmic forceful expulsion of sperm at the urethral meatus. (**a**) and (**b**) = emission; (**c**) = expulsion.

**Figure 2 biology-11-00686-f002:**
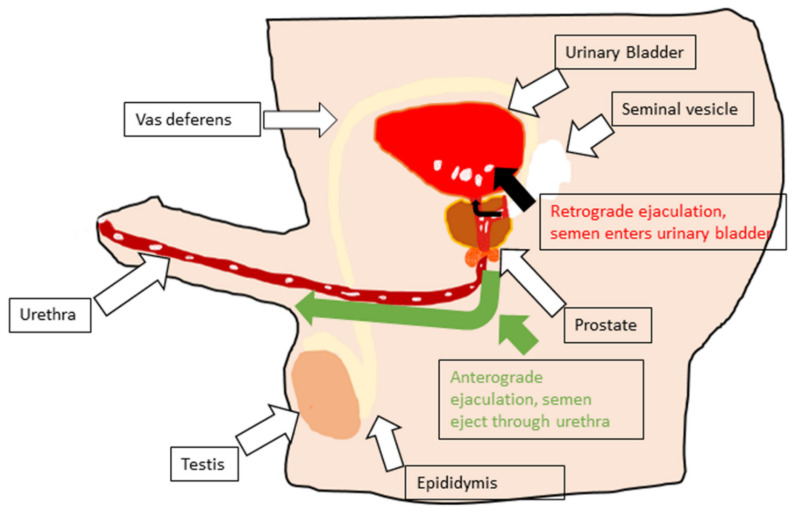
Schematic diagram of organs involved in ejaculation. Anterograde ejaculation (green arrow) and retrograde ejaculation (black arrow) shown in same figure, retrograde ejaculation is due to the inability of the neck of bladder to close completely during expulsion phase. Some amount of semen can be seen in the urinary bladder.

**Figure 3 biology-11-00686-f003:**
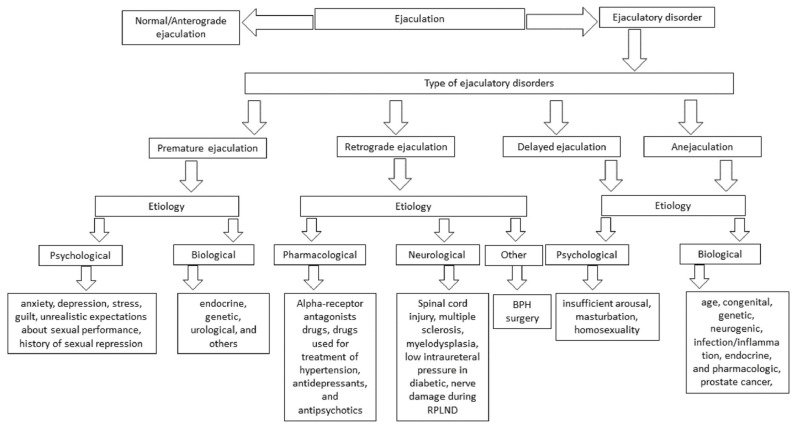
Table showing the normal and abnormal types of ejaculation. Their etiologies are described in brief.

## Data Availability

Not applicable.

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
