# Peer review of "Neurons for Ejaculation and Factors Affecting Ejaculation"

_biology, 2022, doi:10.3390/biology11050686_

Round 1

Reviewer 1 Report

This is a review manuscript describing the anatomical and cellular regulation of ejaculation, including factors involved in ejaculation dysfunction. The review is informative and provides a broad set of topics related to the main focus.

Major comments:

Figure 1 is not particularly informative. If the main point of this figure is to provide anatomical information, then showing the location and interaction of more cells than just Lst neurons (for example the spinal ejaculation generator [SEG] and the three cell populations describe in figure) would be more helpful to the reader than the current two (very similar) panels.

Similarly, the role of the three cell populations shown in figure 2 seems somehow confusing or incomplete. Does the activation of each of those population lead to ejaculation or do they need to be activated as a group or in a specific sequence? This interaction should be explained clearer in the text or in the figure legend.

Some of the factors described in the etiopathogenesis of delayed ejaculation and anejaculation lack significant studies. For example, the citation included for religious factors did not directly test this aspect and it sounds too specific to one religion excluding others. Also, the two studies included for homosexuality do not rigorously determine homosexuality as a factor for ejaculation dysfunction. In fact, one of the studies says in the abstract “there remained no significant effects of sexual orientation on ejaculatory dysfunction”, which is misleading. Similarly for race, the cited study described the higher prevalence in black population more as a trend than an actual contributing factor. My suggestion would be to find better comparative studies showing robust and replicable scientific evidence to support such conclusions. Alternatively, the authors could change their language to describe these aspects more carefully indicating the imprecise nature of such studies, or, just remove them from the list. Lastly, the description of the serotonin reuptake inhibitors for the genetic and pharmacological parts sounds very repetitive.

To improve the impact of the review, the authors could consider the suggestion to include two additional sections. First, a deeper discussion into how the new advances in our understanding of the anatomy underlying ejaculation (section #2 of the manuscript) could help in the long run solve the dysfunctional issues described in #3, #4, and #5 (and if not what are the limitations). Secondly, based on the abstract and title of the manuscript, the authors could expand on the idea that research in animal models needs to be expanded and include a small section describing studies of neuronal regulation of ejaculations in non-mammalian models (for example Drosophila and the Corazonin neurons [Tayler et al., 2012; Zer-Krispil et al., 2018; reviewed in Khan et al., 2021]) or other examples. It may be worth mentioning what the status of our understanding is in mice or other vertebrates (compared to the well-known situation in rats).

Minor comments:

Lines 35-36: It’s unclear what is meant with “a water pressure of 35 more than 5 m”

Lines 79: It’s unclear what is meant with “sexual information”

Lines 93: It’s unclear what is meant with “completely controlled voluntarily”

Line 132: it should probably read “is the most common type of ejaculation dysfunction in men”

In line 245 the authors describe that “Vitamin D causes anxiety, which may be a probable cause of PE in vitamin D deficiency patients”. It should be clarified here if vitamin D deficiency or the treatment is related to anxiety as line 159 says “low anxiety delays ejaculation”.

The reader would benefit from a clarification of the urological organs mentioned in line 251.

Reviewer 2 Report

Dear Authors,

your works is a comprehensive review of Ejaculation in animal models and thus in men, and all of its dysfunction. Nevertheless, style and english should be substantially revised. More images should be welcome, especially regarding ejaculatory dysfunctions. Regarding drugs related to retrograde ejaculation, there are plenty of them, with a large variety of incidence of this dysfunction. Moreover, you forgot a largely known cause of retrograde ejaculation in Urology, which is BPH surgery, thus this paragraph should be implemented. In addition, surgery in prostate cancer is another cause of anejaculation

Reviewer 3 Report

The review is well conducted although the topic is not entirely new.  It represents an overview of the spinal regulation of ejaculation and in my opinion, it could be accepted in the present form. 

Round 2

Reviewer 1 Report

The authors have addressed all my comments and the revised manuscript is much improved.

There were three minor comments that were addressed in the cover letter but not in the revised manuscript.

1)Ques: Lines 35-36: It’s unclear what is meant with “a water pressure of 35 more than 5 m” Ans: We have checked the reference; it is the pressure of prostatic urethra during expulsion.

The units for pressure are usually not meters (m), and the "35" has not units, so it is still confusing to the reader.  So I don't think this sentence is providing accurate information

2) For the responses on Lines 79:  Ans: “sexual information” are “sensory cues prior ejaculation” and Lines 93: “completely controlled voluntarily” Ans: It means the emission phase is avoidable and is not like expulsion phase which is unavoidable.

The author's explanations help understand the concepts much better. These additional explanations should be added to the manuscript to improve the reader's understanding.

3) there are still some grammatical errors that need attention

Reviewer 2 Report

Dear Authors,

implementations to the manuscript improved its quality. Nevertheless, I would put BPH in a differect subsection (not in neurogenic factor, as it would be incorrect), alike other causes or iatrogenic. Other points are all fine
